# Influence of Different Precursors on Content of Polyphenols in *Camellia sinensis* In Vitro Callus Culture

**DOI:** 10.3390/plants12040796

**Published:** 2023-02-10

**Authors:** Maria A. Aksenova, Tatiana L. Nechaeva, Maria Y. Zubova, Evgenia A. Goncharuk, Varvara V. Kazantseva, Vera M. Katanskaya, Petr V. Lapshin, Natalia V. Zagoskina

**Affiliations:** K.A. Timiryazev Institute of Plant Physiology, Russian Academy of Sciences, 127276 Moscow, Russia

**Keywords:** *Camellia sinensis*, *L*-phenylalanine, *trans*-cinnamic acid, naringenin, phenylpropanoids, flavans, proanthocyanidins, lipid peroxidation

## Abstract

Plant tissue cultures are considered as potential producers of biologically active plant metabolites, which include various phenolic compounds that can be used to maintain human health. Moreover, in most cases, their accumulation is lower than in the original explants, which requires the search for factors and influences for the intensification of this process. In this case, it is very promising to use the precursors of their biosynthesis as potential “regulators” of the various metabolites’ formation. The purpose of our research was to study the effect of *L*-phenylalanine (PhA, 3 mM), *trans*-cinnamic acid (CA, 1 mM) and naringenin (NG, 0.5 mM), as components of various stages of phenolic metabolism, on accumulation of various phenolic compound classes, including phenylpropanoids, flavans and proanthocyanidins, as well as the content of malondialdehyde in in vitro callus culture of the tea plant (*Camellia sinensis* L.). According to the data obtained, the precursors’ influence did not lead to changes in the morphology and water content of the cultures. At the same time, an increase in the total content of phenolic compounds, as well as phenylpropanoids, flavans and proanthocyanidins, was noted in tea callus cultures. Effectiveness of precursor action depends on its characteristics and the exposure duration, and was more pronounced in the treatments with PhA. This compound can be considered as the most effective precursor regulating phenolic metabolism, contributing to a twofold increase in the total content of phenolic compounds, flavanes and proanthocyanidins, and a fourfold increase in phenylpropanoids in tea callus cultures.

## 1. Introduction

Plant tissue culture grown in vitro can be potential producers of biologically active substances (BAS) valuable for human health [1,2,3]. Significant progress has been made in plant tissue culture production methods, including ones with a high ability to accumulate these compounds, due to the totipotency property of such cultures, the preservation of intact tissues metabolism specifics and the developed cultivation protocols [4]. It should also be noted that the use of plant tissue cultures makes it possible to obtain BAS regardless of the seasonal factors, under strictly controlled conditions and in short periods of time [5,6].

Among the unique BAS representatives are such secondary metabolites as phenolic compounds (PCs), synthesized in almost all plant cells and exhibiting antioxidant activity [7,8,9]. These metabolites are characterized by the presence of a wide variety of phenolic structural units (from one to several dozen), which is important for their chemical properties and biological activity [10,11]. Currently, significant progress has been made in studying the structure of PCs, their properties and biosynthesis [7,12]. The key PCs precursor is *L*-phenylalanine (PhA), which, with the participation of *L*-phenylalanine ammonia-lyase, is converted into the “first” phenolic metabolite—*trans*-cinnamic acid (CA) (Appendix A). As a result of successive reactions, naringenin (NG) is synthesized from *trans*-cinnamate—an important component of the flavonoid pathway of PCs biosynthesis. The enzymes and genes responsible for these metabolic processes have been well studied [13,14].

PCs are classified as low-molecular-weight antioxidants [9,15,16]. They are able to “extinguish” free radical processes, thereby protecting not only plant cells, but also human cells from the action of reactive oxygen species (ROS) [17]. PCs are effective natural antioxidants that are obtained through the food chain, and play an important role in the human diet as one of the health-saving components [18]. In addition, these plant metabolites are characterized by a variety of pharmacological activities: Capillary strengthening, antimicrobial, anticarcinogenic and cardioprotective [19,20]. The participation of PCs, including flavonoids, in the immune response activation of human cells to infection with coronavirus (SARS-CoV-2) has been shown [21,22]. All this causes great interest in the biotechnological production of plant phenolic metabolome representatives, including using plant tissue cultures.

The accumulation of BAS, including PCs, in plant cells and tissues cultivated in vitro can be regulated by the exposure to various exogenous factors of physical and chemical nature [1,5,23,24]. Use of various plant metabolites precursor compounds may be one of such approaches [25,26]. In the callus of coleus plants (*Coleus blumei*), *L*-tyrosine and PhA increased the number of PCs by 12 and 10 times, respectively [27]. The addition of shikimic acid to the tea plant suspension culture during the stationary growth phase led to an increase in the accumulation of catechins and caffeine in it [28]. Glutathione treatment of the nutrient medium increased the accumulation of PCs, including flavonoids, in *Trifolium resupinatum* cultures [29].

Tea plants (*Camellia sinensis* L.) are among the unique plants characterized by a high ability for PC formation [30,31,32]. Their content in the leaves of young shoots can reach 30% by dry weight. Flavans are the main representatives of the tea phenolic complex, represented by catechins and their oligomeric forms—proanthocyanidins [31,33]. It is known that catechins are characterized by P-vitamin capillary-strengthening activity and are capable of inhibiting angiogenesis and tumor cell invasiveness [34,35]. It was reported that the product obtained from the young shoots—green tea—is characterized by anti-photoaging, stress resistance, and neuroprotective and autophagy properties [33].

Maneuvering with tissue/cell culture techniques, a considerable success has been achieved in the areas of micropropagation and somatic embryogenesis, as well as genetic transformation [36]. There have also been reports about morphophysiological characteristics of tea plant cultures, peculiarities of accumulation and regulation of PCs in them, including flavans [37,38]. Earlier, we noted that when the tea plant cells were introduced in in vitro culture, the ability to produce flavans was preserved, although its biosynthesis was at a lower level than in the intact plant tissues [39,40,41,42]. In this regard, the use of elicitors of various nature (metabolites’ precursors, hormones, microelements, etc.) that contribute to the activation of this process is important.

The purpose of the study was to investigate the effect of the PCs precursors such as *L*-phenylalanine, *trans*-cinnamic acid and naringenin on the callus culture of the tea plant, the PCs accumulation, including phenylpropanoids, flavans and proanthocyanidins, as well as the content of malondialdehyde—as an indicator of the cells stress response to these impacts. This approach will make it possible to find out the prospective of regulating the PCs biosynthesis in cultured in vitro tea cells—as potential producers of pharmacologically valuable plant metabolites.

## 2. Results

### 2.1. Morphological Characteristics and Water Content in Tea Callus Cultures

The tea callus cultures, grown in a liquid culture medium of the basic composition or enriched with various precursors, were dense and had a beige color (Figure 1). However, it is worth noting a darker color of callus under the action of *trans*-cinnamic acid (CA).

The tea callus cultures are characterized by low biomass growth, which in the control was 127 ± 15% by the end of the experiment. When tea callus culture was exposed to *L*-phenylalanine (PhA), it reached 119 ± 13% and in other cases it did not exceed 110 ± 10%. It is worth noting that there is a tendency to a slight decrease in the biomass growth of callus culture in the presence of precursors with respect to the control.

Determination of the water content in tea callus culture, as an indicator of their physiological state, did not reveal considerable differences in the studied variants (Table 1). In all samples, it averaged 92% throughout the entire study period. At the same time, according to the two-way analysis of variance (ANOVA), there was a statistically significant interaction between the precursor treatment and time of cultivation (*p* < 0.035) (Appendix A).

### 2.2. The Total Phenolic Content in Tea Callus Cultures

The total phenolic content (TPC) changed significantly under the action of different PCs precursors, time of cultivation and their combined effect (Appendix A). In the control it was almost equal in cultures of 3-, 7- and 14-days-old, and at the age of 9 days it was almost twice as high (Figure 2). When growing tea callus culture in the medium with PhA, the TPC significantly exceeded the one in control (in most cases by 2 times, except for the 9th day), remaining approximately at the same level throughout the passage.

On the medium containing CA, its maximum accumulation was noted at the beginning of cultivation (day 3), which was 50% higher than the value in the control at the same age. As the tea callus culture continued to grow, the TPC decreased, reaching minimum values at the end of the passage (day 14). Growing callus on a medium with NG was accompanied by activation of the PCs accumulation in them in the first half of the passage relative to control (in 3- and 7-days cultures by 50% and 100%, respectively). In 9- and 14-days calli it was equal to control. The above indicates the accumulation regulating possibility of these secondary metabolites’ representatives for the tea culture in vitro, which depends on the precursor chemical nature and its exposure duration.

### 2.3. The Phenylpropanoids Content in Tea Callus Cultures

There is a statistically significant effect of precursor treatment as well as time of cultivation on the phenylpropanoids content (Appendix A). Its determination revealed that in most of the studied tea callus cultures, namely in control and calluses grown on media with CA and NG, their accumulation had almost equal values, although there was a tendency to higher-level content in the presence of NG (Figure 3).

The exception was tea callus culture exposed to PhA. In this case, the quantity of phenylpropanoids at all stages of growth significantly (at least 400%) exceeded that in the control. Consequently, the presence of PhA in the medium for growing tea callus culture contributed to the accumulation of biogenetically early phenolic metabolism compounds—phenylpropanoids.

### 2.4. The Flavans Content in Tea Callus Cultures

Flavans are the main components of the tea plant phenolic complex [32,43,44]. In the control their content was statistically validly equal during almost the entire growing period (Figure 4). The exception was the 9-day callus, where the number of flavans was higher (by 50%).

This parameter significantly changed under all precursor treatments (*p* < 0.001) and the time of cultivation (*p* = 0.002) (Appendix A). In the tea callus culture exposed to PhA, the accumulation of the metabolites increased by the 7th day (relative to the control by 50%) and remained at this level until the end of the passage. A different trend was observed with the effect of CA on the tea callus culture. The greatest accumulation of flavans in it was on the 3rd and 7th days of growth, exceeding this in the control by an average of 50%. Further, it decreased significantly: By 56% and 80% on day 9 and 14, respectively. In tea callus grown on NG medium, the flavans content exceeded the values of control, starting from the 3rd day of growth (by 50%). By the 7th day, it increased even more and remained at this level until the 9th day, and then dramatically decreased. Thus, there are statistically significant differences in flavans accumulation depending on the combined action of factors (*p* < 0.001) (Appendix A).

An increase in the accumulation of such phenolic metabolism compounds as flavans, the precursor of which is NG, confirms the possibility of its practical use as a “regulator” of their accumulation in tea callus culture.

### 2.5. The Proanthocyanidins Content in Tea Callus Cultures

One of the components of the tea plants flavans complex is their oligomeric and polymeric forms—proanthocyanidins [43,45,46]. These metabolites formation is also characteristic of the callus cultures initiated from these plants’ tissues [47]. As follows from our data, the proanthocyanidins content in the tea callus of control was equal and low at the initial (day 3 and 7) and final (day 14) stages of growth (Figure 5). Only in the 9-day cultures was it high, significantly (3 times) exceeding other indicators.

In cultures grown on medium with PhA, the content of proanthocyanidins in most cases was higher than that in the control: On day 3—by 50%, on day 7—by 200%, remaining at this level until the end of the passage. The exception was a culture at the age of 9 days, in which the number of these metabolites was equal to that of control of the same age. In the tea callus culture grown on the medium with CA, the proanthocyanidins content at all stages of growth was equal, and its high level in the 3-day callus was not statistically reliable. One can even note a great similarity in its accumulation with that in the control (with the exception of the 9-day callus). The presence of NG contributed to the proanthocyanidins accumulation at the initial stage of callus growth and was similar to that of PhA during this period. Its maximum level is noted on the 7th day of growth. With longer exposure to this precursor, the amount of proanthocyanidins decreased: In 9-day cultures by 25% from that of 7-day cultures, and further days it is decreased even more. According to the two-way ANOVA, there were statistically significant effects of the precursor treatments and of the cultivation time, as well as combined effect of these two factors (in all cases *p* < 0.001) (Appendix A).

The data obtained indicate the possibility of regulating the proanthocyanidins accumulation in tea callus cultures through the use of precursors such as PhA and NG.

### 2.6. The Level of Lipid Peroxidation in Tea Callus Cultures

Determination of the malondialdehyde (MDA) content as an indicator of the level of LPO [48] in the callus of control of different ages showed their equal level throughout the passage (Figure 6).

When growing cultures on media with PhA and CA, the LPO level was almost similar to that of the control, although at the last stages of growth (14-day callus) it was slightly lower, which was noted to a greater extent in the variant with the action of CA. A different trend was noted for the tea callus culture grown on the medium with NG. In this case, the 3-day culture had the lowest MDA content and it was 20–30% lower compared to all other variants. By the 7th day, it increased (by about 40%) and remained at this level until the end of the passage. Consequently, the tea callus culture grown on the medium with NG, beginning from the 7th day of cultivation, had a pronounced stress reaction, as evidenced by the content of MDA in it. According to the two-way ANOVA, the statistically significant effect of the precursor treatment was revealed with regard to the level of LPO (*p* < 0.001) (Appendix A).

## 3. Discussion

Tea plants (*Camellia sinensis* L.) are a unique crop of widespread industrial use and are characterized by a high accumulation of phenolic BAS in their young leaves [49,50]. The main components of their phenolic complex are flavans—substances with P-vitamin capillary-strengthening and antimicrobial activity [51,52].

Tea has a limited distribution area, and its ability to form various PCs is high, in this regard, the use of biotechnology methods, including in vitro production of cultured cells and tissues, offers great opportunities for obtaining these BAS of plant origin under strictly controlled conditions. It is known that in most cases cell and tissue cultures are characterized by a lower production capacity with regard to specialized phenolic metabolites; therefore, the search for regulators of this process is important. These may include PCs biosynthesis precursors, such as PhA, CA and NG, which are intermediates of the shikimate, phenylpropanoid and flavonoid pathways [7,12].

### 3.1. Morphological Characteristics and Water Content in Tea Callus Cultures

Plant tissue culture’s morphological characteristics and their changes under the action of various factors allow us to estimate the culture’s state and its response [1,2]. According to our data, the use of precursors did not lead to changes in the tea callus culture morphology and its water content (Figure 1, Table 1).

It can only be noted that when growing tea callus culture on the medium with PhA, at the beginning of cultivation (3 days), its biomass growth was higher compared to the control (by 5%), although these differences were not statistically reliable. Further, after 7 days, the growth of callus slowed down (relative to control by 4–8%). In the *Echinacea purpurea* hairy roots culture, the PhA positive effect on the culture growth was also noted only at the initial stage [53].

### 3.2. The Total Phenolic Content in Tea Callus Cultures

The TPC reflects the ability of tea callus culture to synthesize these secondary metabolites [41,42]. As noted above, it can serve as a criterion for determination the ability of plant cells and tissues to biosynthesize secondary metabolites [41]. According to the data obtained by us, in tea callus cultures grown on the main medium and on media with various precursors, it was PhA that caused rapid activation of their accumulation and the effect remained throughout the cultivation period (Figure 2). The stimulating effect of this precursor on the PCs formation has been repeatedly noted for various plant objects [54,55,56,57].

The regulatory effect of NG on TPC was close to that of PhA on the 7th and 9th day of the tea callus culture growth, but differed on the 3rd and 14th day, when it was lower. These differences may be due to the fact that PhA is the precursor of the initial (shikimate) pathway of PCs biosynthesis, and NG is the precursor of their biogenetically later flavonoid pathway (Appendix A). An increase in the PCs accumulation in the presence of NG was also noted in *Glycine max* seedlings, with the values of the experimental variant exceeding the control ones by 32% [58]. According to our data, CA is considered to be the least “effective” precursor, since its stimulating effect on the TPC was less pronounced compared to PhA and NG, and was manifested only at the initial stage of the tea callus growth. A similar trend was observed in *Larrea divaricata* calluses, in which, in the presence of CA, an increase in the content of these secondary metabolites was also noted only in the first days of exposure, after which it dramatically decreased to control values [59]. When growing a callus culture of *Cichorium intybus* in the Heller culture medium with CA, the TPC also increased [60].

So, these data confirm the thesis about the species-specificity of the higher plants various representative’s reaction, including those cultivated in vitro, to exogenous influences, which include precursors.

### 3.3. The Phenylpropanoids Content in Tea Callus Cultures

Phenylpropanoids are biogenetically early phenolic metabolites of plants, which are characterized by antioxidant activity and attract the attention of researchers [11,12]. The study of their content in tea callus culture grown on the medium with various phenolic precursors showed the greatest effectiveness of PhA in regulating their formation during the entire passaging cycle relative to other compounds, of which one had almost no effect on this process (Figure 3). A significant increase in the accumulation of these metabolites under its action was noted for the callus culture *Larrea divaricata*, *Vitis vinifera* (variety Grenache) and suspension culture *Ocimum tenuiflorum* [56,59,61]. This effect can be explained by the fact that phenylpropanoids are among the simplest representatives of PCs, which are formed at the early stages of biogenesis [7,11]. The presence of PhA in the medium, apparently, contributes to its rapid involvement in the biosynthesis of these metabolites. Thus, as a consequence, there is an increase in phenylpropanoids content in the tea callus culture.

As noted above, the other two precursors of phenolic metabolism had almost no effect on the phenylpropanoids accumulation in tea callus culture. When they were grown on a medium with CA, it remained at the control level almost throughout the entire cultivation period, and by the end of the passage it even decreased below the control values. A decrease in the content of phenylpropanoids, including chlorogenic acid, was also noted in the callus culture of *Cichorium intybus* exposed to CA [60]. In several studies, it has been shown that CA regulates the enzyme activities of phenolic biosynthesis, and the authors speak specifically about the inverse negative relationship between the precursor and enzymes [62,63,64].

As for NG, with its action, there was a slight increase in the phenylpropanoids accumulation during the entire cultivation period relative to control; this is valid for most samples. It is possible that this effect is a consequence of *p*-coumaryl-CoA ligase activity inhibition, which leads mainly to the hydroxy-cinnamic acid accumulation sequentially formed from *p*-coumaric acid [65]. The obtained data reveal a regulatory role of PhA with regard to the increase in the formation of phenylpropanoids in the tea callus culture, in contrast to the other two precursors (CA and NG).

### 3.4. The Flavans Content in Tea Callus Cultures

One of the main tea phenolic complex components are flavans—substances with P-vitamin capillary-strengthening activity [32,66]. The most rapid regulatory effect on their accumulation in tea callus culture was observed in the 3-day callus grown on media with CA and NG, when the amount of these metabolites was 50% higher than in the control (Figure 4). With further growth of these cultures in the variant with CA, it significantly decreased. In the presence of NG, the flavan content in the 7-day callus exceeded that in the 3-day callus, remaining at this level in the 9-day callus, but by the end of the experiment it significantly decreased and was equal to the control.

PhA also contributed to the increase of flavans content in tea callus culture. This effect was observed from the 7th day of cultivation until the end of the passage. At the same time, during this entire period, the quantity of flavans in the callus was equal, that is, it did not depend on the duration of exposure to this precursor. An increase in the accumulation of monomeric forms of flavans was also noted in the leaves of *Triticum aestivum* (“Vánek” variety) after exogenous exposure to PhA [26]. As for the regulatory effect of CA on the flavans accumulation in tea callus culture, compared with other precursors, it was the smallest and manifested only at the initial stages of growth (3rd day). It should also be noted that in most cases, that the content of flavans was equal in the 9-day tea callus cultures, except for the action of CA. This pattern is of interest for our further research.

### 3.5. The Proanthocyanidins Content in Tea Callus Cultures

Proanthocyanidins are oligo- and polymer compounds of flavans [43,45,66]. Their biosynthesis takes place at the final stages of phenolic metabolism (Appendix A). It is worth noting that all the used PCs precursors had a stimulating effect on the proanthocyanidins formation in tea callus culture, the main differences were in the activation period of their accumulation (Figure 5). So, if in the control samples it increased starting from the 9th day of their growth, then with the action of precursors its content enhanced earlier.

Already in the 3-day tea callus cultures, it exceeded the control value by an average of 50% in variants with PhA, CA and NG. In the 7-day callus grown on media with PhA and NG, they were even more pronounced (almost 200%). The high content of proanthocyanidins remained until the end of the growing cycle in the tea callus culture grown on a medium with PhA. Whereas in the tea callus exposed to NG, it gradually decreased. The inducing effect of PhA on the accumulation of flavans’ oligomeric forms was also noted for *Triticum aestivum* [26]. There are no data on the effect of the investigated precursors on the accumulation of proanthocyanidins in plants and cultures in vitro in the literature.

Thus, according to the data obtained, we can say that all precursors contribute to the activation of the proanthocyanidins accumulation already at the early stages of their effect on the tea callus culture.

### 3.6. The Level of Lipid Peroxidation in Tea Callus Cultures

It is known that with the exogenous influence of various factors on plant cells, the activation of ROS formation can occur in them and, as a consequence, the development of oxidative stress [9,15,48].

In our study, the determination of the MDA content, indicating changes in the antioxidant system, in the tea callus culture of the control and experimental variants in most cases did not reveal significant changes to it (Figure 6). After exposure to PhA and CA in the 7- and 9-day cultures, it slightly exceeded that of the control, and in the 14-day cultures it was lower (especially in the variant with the action of CA).

For seedlings of *Fagopyrum esculentum* and *Lens culinaris*, it was shown that in the presence of PhA did not cause changes in the LPO level [67,68]. A similar trend was observed for *Tanacetum parthenium* seedlings, in which CA did not affect the content of MDA [64]. A different picture was noted by us with respect to the effect of NG on the tea callus culture: On the 3rd day of growth, the content of MDA was lower than that of control, whereas in the 7-day cultures this indicator exceeded the control data by 2 times, and this tendency to increase the stress reaction persisted until the end of the experiment. An increased content of MDA after exposure to NG was also noted for *Vigna radiata* seedlings [52].

Based on these data, it can be concluded that only NG had a stressful effect on the tea callus culture. The consequence of this was the activation of the phenolic bioantioxidants accumulation in them, according to the data on the total content of PCs and flavans. This once again confirms the thesis about the important role of flavans in protecting cells from the action of ROS [69].

## 4. Materials and Methods

### 4.1. Plant Material and Experimental Conditions

The object of the study was a heterotrophic callus culture obtained from the stem of young shoots of a tea plant (*Camellia sinensis* L., Georgian variety). For its cultivation, Heller culture medium containing 5 mg/L 2,4-dichlorophenoxyacetic acid, 25 g/L glucose and 7 g/L agar was used [47]. The calluses were grown under the conditions of the growth cabinet at the IFR RAS (25 °C, relative air humidity 70%, darkness 24 h). The duration of the subcultivation was 40 days.

When setting up the experiment, callus tissues (the weight of each callus was 220–250 mg) were placed in conical flasks (100 mL) containing the liquid Heller culture medium (15 mL) of the basic composition (control) or enriched with PhA (3 mM), CA (1 mM) or NG (0.5 mM) (Appendix A). The precursors were dissolved in distilled water, sterilized through membrane filters (MILLEX GV, 0.22 µm, Merck, Darmstadt, Germany) and added to the autoclaved liquid Heller culture medium. Precursors’ concentrations were selected in preliminary experiments. The tea callus cultures were grown under conditions in the growth cabinet on a shaker at a frequency of 90 rpm (25 °C, relative air humidity 70%, darkness 24 h) and analyzed on 3rd, 7th, 9th and 14th days. For biochemical studies, callus tissues were frozen with liquid nitrogen and stored at −70 °C.

### 4.2. Determination of Morphological Characteristics of Tea Callus Cultures

The main indicators in estimating the morphological characteristics of tea callus cultures were their appearance, color and texture. The increase in weight of calluses was determined considering their raw mass at the beginning and at the end of the experiment, expressing this indicator as a percentage.

### 4.3. Determination of Water Content in Tea Callus Cultures

The tea callus was dried in the thermostat BD-115 (Binder, Tuttlingen, Germany) to a constant weight (70 °C, 48 h). Calculation of water amount was carried out using the standard formula [70].

### 4.4. Extraction of Phenolic Compounds from Tea Callus Culture

Samples of frozen in liquid nitrogen tea callus culture (50 mg) were ground in a porcelain mortar, extracted with 96% ethanol [71]. The homogenate was kept at 45 °C (thermostat Gnom, Moscow, Russia) for 30 min; it was centrifuged at 16,000× *g* for 5 min (centrifuge Minispin, Göttingen, Germany). The sediment was separated and the supernatant fraction was used to determine various phenolic compounds by spectrophotometric methods.

### 4.5. Determination of Different Phenolic Compounds Classes in Tea Callus Culture

The determination of the total phenolic content quantity was carried out with the Folin–Chokalteu reagent at 725 nm [72,73]. The content of phenylpropanoids was analyzed by direct spectrophotometry of solutions at 330 nm [74]. The amount of flavans was determined with vanillin reagent at 500 nm [75] and proanthocyanidins, with butanol reagent at 550 nm [47,76].

The total phenolic content was expressed in mg-eq. gallic acid/g dry weight, the content of phenylpropanoids in mg-eq. caffeic acid/g dry weight, the content of flavans in mg-eq. epicatechin/g dry weight, the content of proanthocyanidins in mg-eq. cyanidine/g dry weight.

### 4.6. Determination of the Level of Lipid Peroxidation in Tea Callus Culture

The level of lipid peroxidation in callus tissues was assessed by the content of malonic dialdehyde (MDA), using a reaction with thiobarbituric acid, the optical density of the solution was measured at 532 nm [77]. To calculate the MDA content (µmol g^−1^ dry weight), a molar extinction coefficient equal to 1.56 × 10^−5^ cm^−1^ M^−1^ was used [47].

###  4.7. Statistical Analysis

All determinations were carried out in three biological and three analytical replicates. The obtained data were statistically processed using Microsoft Excel 2010 14.0 (Redmond, WA, USA) and SigmaPlot 12.2 (Technology Networks, Sudbury, UK) software. The figures show the arithmetic means ± standard deviations (±SDs). Statistical analyses of data were performed using two-way analysis of variance (ANOVA). Mean separation was performed using Normality Test (Shapiro–Wilk) and all Pairwise Multiple Comparison Procedures (Tukey test). The significant differences at *p* < 0.05 are denoted by different Latin letters: Uppercase letters indicate significant differences between precursor treatments in alphabetic order from highest to lowest, lowercase letters indicate significant differences between duration of precursor treatments in alphabetic order from highest to lowest.

## 5. Conclusions

Tea plants (*Camellia sinensis* L.) are of great interest, and are widely used by the population of our planet in the daily diet and are important for human health. A characteristic feature of this culture is the accumulation of various PCs, including flavans—compounds with P-vitamin capillary-strengthening activity. The limited tea growth in natural conditions and its unique properties serve as important factors for its cultivation in vitro, where, as previously shown, the ability of PCs formation of the initial explants was preserved. At the same time, its quantity is lower, which requires the search for factors and influences regulating this process.

We have investigated the influence of PCs biosynthesis precursors, namely PhA, CA and NG, on the tea callus cultures and on the accumulation of phenolic compounds in them. All these substances did not affect the morphology, water content or biomass growth of cultures. In all cases, these data were close to those of the control. At the same time, the cultivation of tea calluses on media with precursors was accompanied by a change in the accumulation of phenolic compounds in them. This effect depended on the nature of the precursor and the duration of its exposure. Thus, when growing calluses on a medium with PhA, the content of all the main PCs classes increased, but this was most evident at the level of biogenetically early metabolites—phenylpropanoids. This effect persisted throughout the entire period of their growth. Consequently, the addition of the PhA to the Heller culture medium contributes to the activation of phenolic metabolism in callus tea cultures and the accumulation of biologically active phenolic bioantioxidants in them.

When tea calluses were exposed to CA, the PCs accumulation increased only in the first days of exposure. This effect is due to an increase in the level of flavans and proanthocyanidins. As for the regulatory effect of NG, after its exposure to tea callus culture, the PCs accumulation, including flavans and proanthocyanidins, increased, although this effect is probably due to its stress effect. The content of phenylpropanoids in them was almost at the level of control and did not change during the entire growth cycle. Hence, the action of naringenin, as a precursor of the flavonoid pathway of biosynthesis of phenolic compounds (Appendix A), contributed to the accumulation of metabolites synthesized at the final stages of this process, namely flavanols of various degrees of polymerization.

Consequently, various precursors can be used to regulate directly the PCs’ various classes accumulation for in vitro plant cell cultures, in particular tea callus culture, which is important for obtaining valuable biologically active substances of plant origin by biotechnological methods.

## Figures and Tables

**Figure 1 plants-12-00796-f001:**
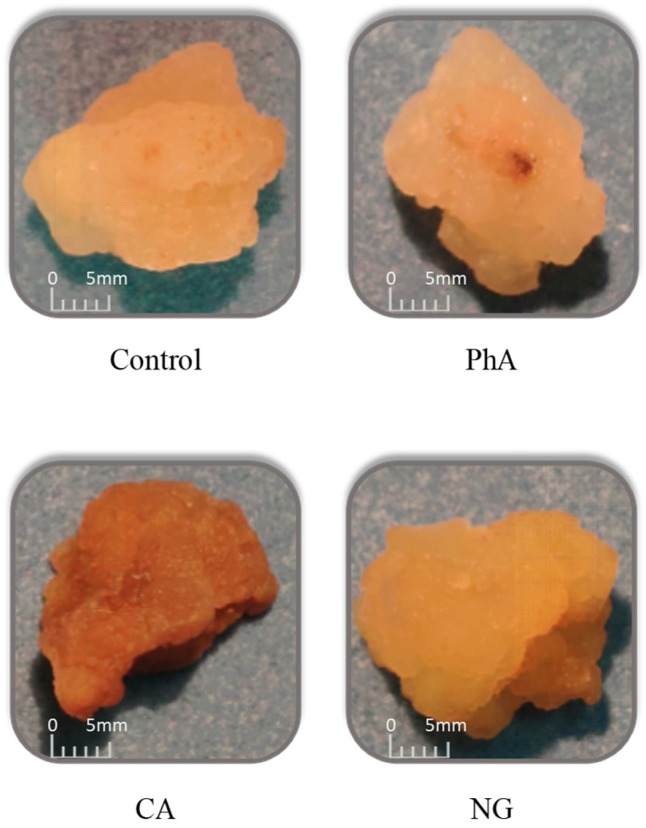
The tea callus cultures grown on the liquid Heller culture medium of the basic composition (Control) or enriched with *L*-phenylalanine (PhA; 3 mM), *trans*-cinnamic acid (CA; 1 mM) or naringenin (NG; 0.5 mM). Callus culture age—9 days.

**Figure 2 plants-12-00796-f002:**
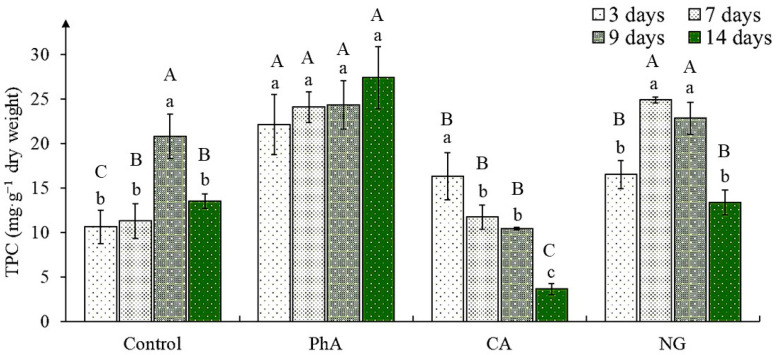
The total phenolic content (TPC) in tea callus cultures of various ages grown on the liquid Heller culture medium of the basic composition (Control) or enriched with *L*-phenylalanine (PhA; 3 mM), *trans*-cinnamic acid (CA; 1 mM) or naringenin (NG; 0.5 mM). Results are expressed as means ± SDs, *n* = 3. The significant differences at *p* < 0.05 are denoted by different Latin letters: Uppercase letters indicate significant differences between precursor treatments in alphabetic order from highest to lowest, lowercase letters indicate significant differences between duration of precursor treatments in alphabetic order from highest to lowest. Pairwise multiple comparisons were carried out using Tukey’s range test.

**Figure 3 plants-12-00796-f003:**
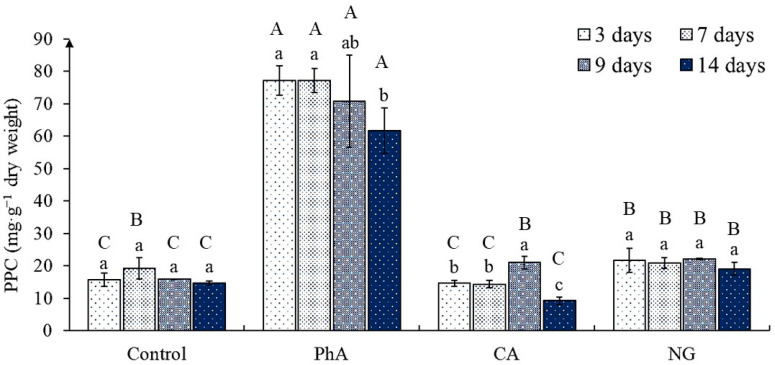
The phenylpropanoids content (PPC) in tea callus cultures of various ages grown on the liquid Heller culture medium of the basic composition (Control) or enriched with *L*-phenylalanine (PhA; 3 mM), *trans*-cinnamic acid (CA; 1 mM) or naringenin (NG; 0.5 mM). Results are expressed as means ± SDs, *n* = 3. The significant differences at *p* < 0.05 are denoted by different Latin letters: Uppercase letters indicate significant differences between precursor treatments in alphabetic order from highest to lowest, lowercase letters indicate significant differences between duration of precursor treatments in alphabetic order from highest to lowest. Pairwise multiple comparisons were carried out using Tukey’s range test.

**Figure 4 plants-12-00796-f004:**
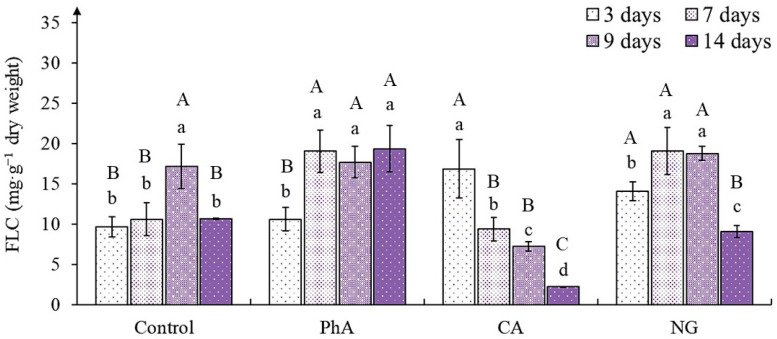
The flavans content (FLC) in tea callus cultures of various ages grown on liquid Heller culture medium of the basic composition (Control) or enriched with *L*-phenylalanine (PhA; 3 mM), *trans*-cinnamic acid (CA; 1 mM) or naringenin (NG; 0.5 mM). Results are expressed as means ± SDs, *n* = 3. The significant differences at *p* < 0.05 are denoted by different Latin letters: Uppercase letters indicate significant differences between precursor treatments in alphabetic order from highest to lowest, lowercase letters indicate significant differences between duration of precursor treatments in alphabetic order from highest to lowest. Pairwise multiple comparisons were carried out using Tukey’s range test.

**Figure 5 plants-12-00796-f005:**
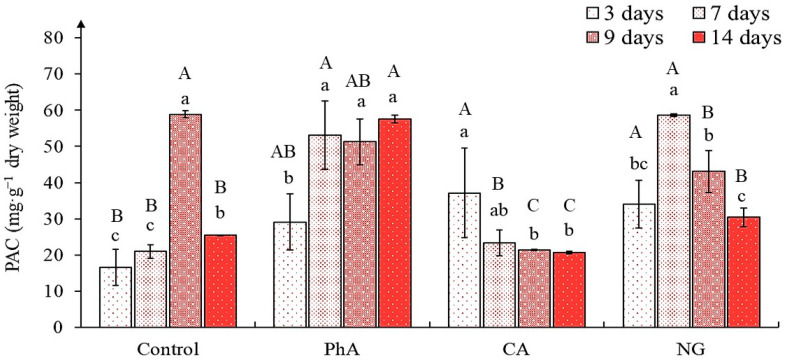
The proanthocyanidins content (PAC) in tea plant callus cultures of various ages grown on the liquid Heller culture medium of the basic composition (Control) or enriched with *L*-phenylalanine (PhA; 3 mM), *trans*-cinnamic acid (CA; 1 mM) or naringenin (NG; 0.5 mM). Results are expressed as means ± SDs, *n* = 3. The significant differences at *p* < 0.05 are denoted by different Latin letters: Uppercase letters indicate significant differences between precursor treatments in alphabetic order from highest to lowest, lowercase letters indicate significant differences between duration of precursor treatments in alphabetic order from highest to lowest. Pairwise multiple comparisons were carried out using Tukey’s range test.

**Figure 6 plants-12-00796-f006:**
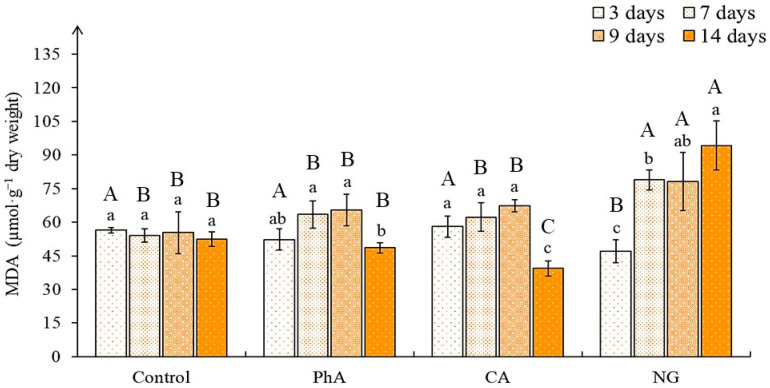
The malondialdehyde (MDA) content in tea callus cultures of various ages grown on the liquid Heller culture medium of the basic composition (Control) or enriched with *L*-phenylalanine (PhA; 3 mM), *trans*-cinnamic acid (CA; 1 mM) or naringenin (NG; 0.5 mM). Results are expressed as means ± SDs, *n* = 3. The significant differences at *p* < 0.05 are denoted by different Latin letters: Uppercase letters indicate significant differences between precursor treatments in alphabetic order from highest to lowest, lowercase letters indicate significant differences between duration of precursor treatments in alphabetic order from highest to lowest. Pairwise multiple comparisons were carried out using Tukey’s range test.

**Table 1 plants-12-00796-t001:** Water content in tea callus cultures at different ages grown on the liquid Heller culture medium of the basic composition (Control) or enriched with *L*-phenylalanine (PhA, 3 mM), *trans*-cinnamic acid (CA, 1 mM) or naringenin (NG, 0.1 mM).

Variants	Water Content in Cultures of Different Ages, %			
	3 Days	7 Days	9 Days	14 Days
Control	92.29 ± 0.83 ^Aa^	92.13 ± 0.02 ^Ba^	92,65 ± 0.19 ^Aa^	91.39 ± 0.21 ^Ba^
PhA	91.20 ± 0.23 ^Bc^	92.05 ± 0.25 ^Bb^	92,10 ± 0.27 ^Bb^	93.17 ± 0.64 ^Aa^
CA	92.33 ± 0.20 ^Aa^	92.42 ± 0.94 ^Ba^	92,10 ± 0.52 ^ABa^	93.02 ± 0.50 ^Aa^
NG	91.69 ± 0.15 ^Ab^	94.01 ± 0.23 ^Aa^	91,65 ± 0.81 ^ABb^	92.21 ± 0.39 ^Ab^

Results are expressed as means ± standard deviations, *n* = 3. The significant differences at *p* < 0.05 are denoted by different Latin letters: Uppercase letters indicate significant differences between precursor treatments in alphabetic order from highest to lowest, lowercase letters indicate significant differences between duration of precursor treatments in alphabetic order from highest to lowest. Pairwise multiple comparisons were carried out using Tukey’s range test.

## Data Availability

The data presented in this study are available on request from the corresponding author. Data are contained within the article or Appendix A.

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
