# Peer review of "Influence of Different Precursors on Content of Polyphenols in Camellia sinensis In Vitro Callus Culture"

_plants, 2023, doi:10.3390/plants12040796_

Round 1
Reviewer 2 Report
Authors conducted experiments with different precursors added to the culture medium, in order to associate with phenolic compounds production in calli culture of camellia sinensis. The effects of precursors is knowledge and there are few novelties associated with the paper. In addition, there are a lot of changes required for better presentation of paper, including changes in statystical analysis, which authors should be revised. Thus, the paper should be extensive revisions before a consideration for publication in Plants. Detailed review and suggestions of changes are added directly in PDF.

Round 2
Reviewer 2 Report
Authors provide some superficial corrections in the manuscript, but failed to response to the most important queries (e.g. statystical analysis) as recommended in the first review. Also, since superficial corrections are not did in completely (see item 3. Discission, instead of Discussion). Thus, I return the paper with some additional comments in this last version, but authors can revise this and review one version with several appointments before a resubmission. I believe that this paper has potential for publication in plants, but at the same time, authors don't provide a strong review as requested previously in review one. In this way, I believe that paper could be rejected and resubmitted to the journal after a strong review by authors.

Author Response
Dear Reviewer,
We are thankful for your consideration regarding our manuscript titled “Influence of different precursors on content of polyphenols in callus of Camellia sinensis” (“Regulation by precursors (L-phenylalanine, trans-cinnamic acid, naringenin) of polyphenol accumulation in Camellia sinensis in vitro cultures”). We are grateful to you for the detailed comments of our manuscript. Most of them have been taken into account and corrected. Answers and edits to the article are indicated in an additional PDF-file.
We also would like to highlight the main points related to your comments and suggestions:
- We have taken into consideration the recommendations for the Abstract section and made the appropriate changes.
- We used the term "Tea callus culture" in the manuscript, based on your proposal to standardize this terminology.
- As for the morphology of tea callus cultures, the main criteria for their evaluation have been corrected by us.
- As regards the statistical analysis of the results, we calculated all the data in the program Microsoft Excel 2010 14.0 (Redmond, W.A., USA) and SigmaPlot 12.2 (Technology Networks, Sudbury, UK) software. The significance of differences in the mean values was determined by Tukey’s range test at p ≤ 0.05 and denoted by different Latin letters. In addition, in the Supplement section, we presented figures where uppercase letters indicate significant differences between duration of precursors treatments in alphabetic order from highest to lowest, lowercase letters indicate significant differences between precursors treatments in alphabetic order from highest to lowest, all the significant differences were determined by Tukey’s range test (p ≤ 0.05).
Our main aim was to compare the effectiveness of the precursors on the formation of phenolic metabolites in callus cultures of tea. This approach is of great importance when using precursors to obtain biologically active substances based on plant cell cultures.
- As for the question on the section Materials and methods, it was shortened according to the recommendation of the editorial office of the journal
- The Conclusion section has been corrected in accordance with your comments.
Thank you again for your careful work with our manuscript, remarks, and comments on it.
Team of authors
